# Rapid Evaporative Ionization Mass Spectrometry-Based Lipidomics for Identification of Canine Mammary Pathology

**DOI:** 10.3390/ijms231810562

**Published:** 2022-09-12

**Authors:** Domenica Mangraviti, Jessica Maria Abbate, Carmelo Iaria, Francesca Rigano, Luigi Mondello, Marco Quartuccio, Fabio Marino

**Affiliations:** 1Department of Chemical, Biological, Pharmaceutical and Environmental Sciences, Polo Universitario Papardo, University of Messina, 98166 Messina, Italy; 2Department of Veterinary Sciences, Polo Universitario Annunziata, University of Messina, 98168 Messina, Italy; 3Chromaleont s.r.l., c/o, Department of Chemical, Biological, Pharmaceutical and Environmental Sciences, University of Messina, 98168 Messina, Italy; 4Unit of Food Science and Nutrition, Department of Medicine, University Campus Bio-Medico of Rome, 00128 Rome, Italy

**Keywords:** REIMS, iKnife, Rapid Evaporative Ionization Mass Spectrometry, lipidomic, canine mammary pathology, CMT, canine mammary tumors, veterinary surgery

## Abstract

The present work proposes the use of a fast analytical platform for the mass spectrometric (MS) profiling of canine mammary tissues in their native form for the building of a predictive statistical model. The latter could be used as a novel diagnostic tool for the real-time identification of different cellular alterations in order to improve tissue resection during veterinary surgery, as previously validated in human oncology. Specifically, Rapid Evaporative Ionization Mass Spectrometry (REIMS) coupled with surgical electrocautery (intelligent knife—iKnife) was used to collect MS data from histologically processed mammary samples, classified into healthy, hyperplastic/dysplastic, mastitis and tumors. Differences in the lipid composition enabled tissue discrimination with an accuracy greater than 90%. The recognition capability of REIMS was tested on unknown mammary samples, and all of them were correctly identified with a correctness score of 98–100%. Triglyceride identification was increased in healthy mammary tissues, while the abundance of phospholipids was observed in altered tissues, reflecting morpho-functional changes in cell membranes, and oxidized species were also tentatively identified as discriminant features. The obtained lipidomic profiles represented unique fingerprints of the samples, suggesting that the iKnife technique is capable of differentiating mammary tissues following chemical changes in cellular metabolism.

## 1. Introduction

Mammary gland pathology is a common health issue in intact female dogs, and careful differentiation between neoplastic and non-neoplastic conditions is critical, influencing patient treatment and prognosis. Most mammary gland specimens undergoing histological examination are neoplastic, and canine mammary tumors (CMT) are increasingly detected, reaching an incidence and malignant potential similar to those of breast cancer in women [1,2,3,4]. However, non-neoplastic mammary enlargement and masses of benign origin may also occur, including mastitis, hyperplasia and cystic lesions [1], and their differentiation from cancer based on their macroscopic appearance has not been demonstrated. Pathological examination (e.g., cytology and histology) is necessary to achieve diagnosis [5,6], and histopathology still remains the gold standard diagnostic method for accurate classification and grading of canine mammary tumors, also providing essential prognostic information [2].

Conservative surgery is considered the treatment of choice for canine mammary tumors, occasionally in combination with chemotherapy and radiotherapy [3]. The veterinary surgeon must precisely locate the tumor extent, merging macroscopic criteria, imaging studies and background knowledge concerning the type of malignancy. Histological assessment of surgical margins is a useful routine approach to predict local tumor recurrence [2] and requires multiple steps over several days. Furthermore, although the resected margins may be histologically unsuspecting, up to 58% of dogs are referred with a new neoplasm in the remaining mammary tissue after the first surgery [7], and re-operation is cost-inefficient and leads to physical morbidity. Therefore, more accurate and less time-consuming diagnostic methods are needed as supportive procedures in a clinical setting.

In recent years, numerous studies have been directed toward the characterization of biological tissues, including an intensive search for new metabolic biomarkers [8,9]. Above all, tissue lipidomics is gaining increasing interest as comprehensive profiling of lipids, which allows for studying lipid pathways and a better understanding of their changes under different pathological conditions [10,11,12,13,14]. Due to their biological functions in cellular processes, lipids are closely involved in several organic disorders, including inflammation and cancer, reflecting essential morpho-functional cellular adaptations [10,12,15]. Significant advances have been made in mass spectrometric-based methods for the identification of biological tissues, and new technologies capable of accurately profiling and quantifying lipid species for use as metabolic markers in the biomedical field are of growing scientific interest.

Rapid Evaporative Ionization Mass Spectrometry (REIMS) was developed for real-time detection of metabolites useful for intraoperative tissue identification, merging classic surgical electrocautery and mass spectrometry (MS), resulting in the so-called iKnife technique [16,17,18,19,20,21], which means intelligent knife. Such an approach exploits the concept of machine learning since MS data are collected in a database, usable for the rapid identification of unknown samples through the employment of statistical tools. During tissue dissection, the aerosols produced by the iKnife are analyzed by an MS system, which provides information on the lipid composition of biological tissues and investigates the differences in lipid species and relative concentrations between different tissues [22]. To date, the REIMS technology has been successfully employed for metabolic phenotyping tumors and to discriminate from healthy tissues in different anatomical sites, including the brain, breast, uterus and colon, exhibiting high diagnostic accuracy, with the rapid availability of the data as a major resource [20,23,24,25,26]. Of note, REIMS technology has been validated in clinical research for the diagnosis of biological tissue intraoperatively in human surgical oncology, significantly improving intraoperative margin control and surgical outcomes [25,26,27,28,29].

The ability to rapidly identify and classify biological tissues based on their metabolic phenotypes using a REIMS approach could represent an important advantage over routine methods in monitoring tissue resection in the field of veterinary surgical oncology. Therefore, this study aimed to investigate the performance of the iKnife technology in metabolic profiling pathological canine mammary tissues and differentiate them from healthy mammary gland samples based on different metabolic phenotypes using histologically validated ex vivo samples. To the best of the authors’ knowledge, no studies have previously investigated ambient mass spectrometry-based analysis of lipids for rapid identification of canine mammary pathology.

## 2. Results

### 2.1. Mammary Samples

A total of thirty-nine mammary samples from 23 female dogs were collected and used for mass spectrometric investigations. The histopathological diagnosis classified our samples in normal (*n* = 12), hyperplastic/dysplastic (*n* = 6), inflammatory (*n* = 7) and cancerous (*n* = 14). In more detail, hyperplastic/dysplastic mammary samples included one regular lobular hyperplasia, four lobular hyperplasias with fibrosis and three cystic dilations of mammary ducts. A moderate to severe, chronic, multifocal to coalescing lymphoplasmacytic and histiocytic mastitis was observed in all inflamed mammary gland samples included in the study. Canine mammary tumors included six benign neoplasms (*n* = 2 tubulopapillary adenoma; *n* = 3 complex adenoma; *n* = 1 intraductal papillary adenoma) and eight malignant tumors (*n* = 2 complex carcinoma, grade 1; *n* = 2 tubular carcinoma, grade 1; *n* = 2 tubulopapillary carcinoma; grade 2; *n* = 2 comedocarcinoma, grade 3).

### 2.2. Database Building and Generation of a Classification Model

Mass spectrometric data generated from histologically validated samples (*n* = 20; training samples) and representative for each histological type were employed to build the internal database and the four classes classification model, further validated by performing two *in silico* tests. In particular, the classification model reflects the histopathological classification of mammary samples (normal; hyperplastic/dysplastic; inflammatory; tumors). Initially, principal component analysis (PCA) was applied to point out similarities and differences in the data set, considering the mass range of 200–1000 *m*/*z*. The first two principal components (PCs) explain 53.1% of the total variance. An appropriate separation was observed between the scores of healthy mammary samples (blue) vs. other groups (green, yellow and red) relating to PC1 (30% of variance), as depicted in Figure 1.

Then, a supervised Linear Discriminant Analysis (LDA) analysis was applied to “force” the same pathological conditions to occupy closer positions in the multidimensional statistical space while maximizing the distance between different conditions. Three-dimensional (3D) visualization of the resulting PCA/LDA model clearly shows the net demarcation between healthy (blue) and other classes of mammary samples (Figure 2A). In particular, the 2D visualizations highlight the well-defined separation between inflamed tissues (green), normal (blue) and other conditions (i.e., hyperplasia/dysplasia; tumors) along LD2 (Figure 2B), as well as between neoplasms (red) and hyperplastic/dysplastic mammary tissues (yellow) along LD3 (Figure 2C), which are quite overlapped along LD1 and LD2 as shown in Figure 2B.

The model was validated by applying two different *in silico* approaches. Specifically, the 5-fold cross-validation showed an overall classification accuracy of 94.76% and a very low percentage of failures and outliers (Appendix A).

Noteworthy, tumors were never wrongly identified as healthy mammary glands or inflammatory samples; only in three cases were they identified as hyperplasia/dysplasia, while all healthy tissue and hyperplastic/dysplastic tissues were correctly identified, demonstrating the specificity of 100%. Similarly, one file-out validation showed an overall incorrect rate of about 10%, which mostly regarded misidentification of pathological states but never misclassification of neoplasms as healthy tissue or inflammation (Appendix A).

Moreover, in this case, no failures were registered for healthy tissues. In order to test the recognition capability of REIMS technology, a number of mammary samples (*n* = 19) with known histological diagnoses that were representative of each class and not previously included in the statistical model were submitted to the recognition step in real-time. The software returned, in a few seconds, the output of the analysis performed, revealing that all tested samples were correctly identified with a similarity rate above 98%. In particular, all healthy mammary samples were recognized with 100% similarity (Figure 3A), and supervised analysis reduced differences among neoplastic samples, allowing for properly identifying the tumors reflecting histopathological diagnosis (Figure 3B). Finally, also in cases of inflammatory (Figure 3C) and hyperplastic/dysplastic tissues, the model returned proper feedback (Figure 3D).

### 2.3. Determination of Discriminant Features 

All REIMS-TOF spectra acquired in the mass range 100–2000 *m*/*z* were processed as reported in paragraph 4.3.3. for background subtraction and lock mass correction (255.233 *m*/*z*). The negative ionization mode was chosen for data acquisition considering rich molecular information and high intensities signals observed in the spectra of biological tissues, consisting of deprotonated lipid molecules or sodium adduct ([M-H]^−^, [M-2H]^−^, [M+Na-2H]^−^), as well as demethylated ([M-CH_3_]^−^) and dehydrated ([M-H_2_O-H]^−^) lipid molecules and chloride adducts ([M+Cl]^−^), as in Table 1 and Table 2.

Moreover, the REIMS ionization mechanism was already reported to favor the detection of lipid molecules against other macromolecules present in the biological tissues (e.g., proteins) [30]. As a consequence, the totality of identified compounds belongs to the lipid class.

**Table 1 ijms-23-10562-t001:** List of discriminant features according to the PC1 loading plot, along with the calculation of the mass error (Δ) resulting from the difference between the theoretical mass (Exact Mass) and the observed *m*/*z* value, and a note about the statistical class in which it was mainly detected and/or quantitative comparison. * oxidized species; # confirmed by the literature [31]; + confirmed by the literature [24]; † confirmed by the literature [26,28]; $ confirmed by the literature [32]. The assigned compounds indicated with symbols (#,+,†,$) were already reported in previous studies dealing with cancer in both animal and human tissues.

Assigned Compound	Lipid Class	Detected Ion	*m*/*z*	Exact Mass	Δ	Note
Palmitoleic acid (C16:1)	Fatty acid	[M-H]^−^	253.2167	253.2173	2.37	Healthy
Palmitic acid (C16:0)	Fatty acid	[M-H]^−^	255.2324	255.2324	0.00	Tumor/Hyperplasia/Dysplasia > Healthy/Mastitis
Oleic acid (C18:1)	Fatty acid	[M-H]^−^	281.2480	281.2480	0.00	Healthy > Hyperplasia/Dysplasia > Tumor > Mastitis
Stearic acid (C18:0)	Fatty acid	[M-H]^−^	283.2636	283.2643	2.47	Tumor/Hyperplasia/Dysplasia > Healthy/Mastitis
Arachidonic acid (C20:4)	Fatty acid	[M-H]^−^	303.2333	303.2330	0.99	Hyperplasia/Dysplasia/Tumor > Mastitis
Docosatetraenoic acid (C22:4)	Fatty acid	[M-H]^−^	331.2646	331.2643	0.91	Tumor
Lyso-PA(C16:0)	Phospholipid	[M-H_2_O-H]^−^	391.2240	391.2250	2.56	Tumor > Hyperplasia/Dysplasia/Mastitis
PG(C36:1-2OH) *	Phospholipid	[M-2H]^2−^	403.2597	403.2660	15.62	Mastitis > Tumor/Hyperplasia/Dysplasia
Lyso-PA(C18:1)	Phospholipid	[M-H_2_O-H]^−^	417.2397	417.2406	2.16	Tumor/Hyperplasia/Dysplasia/Mastitis
Lyso-PA(C18:0)	Phospholipid	[M-H_2_O-H]^−^	419.2555	419.2563	1.91	Tumor/Mastitis > Hyperplasia/Dysplasia/Healthy
Lyso-PA(C20:4)	Phospholipid	[M-H_2_O-H]^−^	439.2247	439.2250	0.68	Tumor/Hyperplasia/Dysplasia/Mastitis
Cer(C34:1) ^#^	Sphingolipid	[M+Cl]^−^	572.4811	572.4815	0.70	Hyperplasia/Dysplasia > Tumor/Mastitis > Healthy
PA(C34:1) ^+†^	Phospholipid	[M-H]^−^	673.4815	673.4814	0.15	Tumor/Mastitis
SM(d18:1/16:0) ^+^	Sphingolipid	[M-CH_3_]^−^	687.5432	687.5334	3.35	Tumor/Hyperplasia/Dysplasia > Mastitis > Healthy
PA(C36:2) ^+†^	Phospholipid	[M-H]^−^	699.4948	699.4970	3.15	Tumor/Mastitis > Hyperplasia/Dysplasia > Healthy
PE(C34:0) PE(O-C34:1) PC(C32:0)	Phospholipid	[M-H]^−^ [M-H]^−^ [M-CH_3_]^−^	718.5389	718.5392	0.42	Tumor/Hyperplasia/Dysplasia/Mastitis
PA(C38:4)	Phospholipid	[M-H]^−^	723.4965	723.4970	0.69	Mastitis > Tumor/Hyperplasia/Dysplasia
PE(O-C36:3)	Phospholipid	[M-H]^−^	726.5252	726.5443	13.90	Healthy > Tumor/Hyperplasia/Dysplasia/Mastitis
PG 32:0;O *	Phospholipid	[M-H]^−^	737.4964	737.4974	1.36	Tumor/Mastitis > Hyperplasia/Dysplasia
PE(C36:1) ^$†^ PC(C34:1) ^+†^	Phospholipid	[M-H]^−^ [M-CH_3_]^−^	744.5546	744.5549	0.40	Tumor > Hyperplasia/Dysplasia > Mastitis
PE(O-C38:6)	Phospholipid	[M-H]^−^	748.5208	748.5287	10.55	Mastitis > Tumor
PE(O-C38:5) PC(O-36:5) PE(P-38:4) ^+$^	Phospholipid	[M-H]^−^ [M-CH_3_]^−^ [M-H]^−^	750.5417	750.5443	3.46	Tumor/Hyperplasia/Dysplasia/Mastitis > Healthy
PA(C40:4)	Phospholipid	[M-H]^−^	751.5316	751.5283	4.39	Tumor/ Hyperplasia/Dysplasia/ Mastitis > Healthy
PE(C38:4) ^†$^ PC(C36:4) ^†^	Phospholipid	[M-H]^−^ [M-CH^3^]^−^	766.5389	766.5392	0.39	Tumor/Hyperplasia/Dysplasia/Mastitis
PE(C40:4)	Phospholipid	[M-H]^−^	794.5692	794.5705	1.64	Tumor/Hyperplasia/Dysplasia
PI(C38:4) ^#+$^	Phospholipid	[M-H]^−^	885.5492	885.5499	0.79	Tumor/Mastitis
TG(C52:3) ^†^	Triglycerid	[M+Cl]^−^	891.7150	891.7214	7.83	Healthy
TG(C52:2) ^†^	Triglycerid	[M+Cl]^−^	893.7300	893.7370	6.76	Healthy
TG(C54:4) ^†^	Triglycerid	[M+Cl]^−^	917.7308	917.7370	9.57	Healthy
TG(C54:3) ^†^	Triglycerid	[M+Cl]^−^	919.7439	919.7527	17.14	Healthy
TG(C54:2) ^†^	Triglycerid	[M+Cl]^−^	921.7525	921.7683	0.79	Healthy

**Table 2 ijms-23-10562-t002:** List of discriminant features according to the PC2 loading plot, along with the calculation of the mass error (Δ) resulting from the difference between the theoretical mass (Exact Mass) and the observed *m*/*z* value, and a note about the statistical class in which it was mainly detected and/or quantitative comparison. * oxidized species. For the *m*/*z* value 389.1940, different compounds (keto, hydroxyl, epoxy), characterized by the same exact mass and all belonging to the prostaglandin class, were generated into the humane metabolome database. Similarly, the oxidized phosphatidylglycerol phosphate (PGP) species can contain different oxidation products of arachidonic acid, including prostaglandins. ^+^ confirmed by the literature [24]; ^†^ confirmed by the literature [26,28].

Assigned Compound	Lipid Class	Detected Ion	*m*/*z*	Exact Mass	Δ	Note
Linoleic acid (C18:2)	Fatty acid	[M-H]^−^	279.2326	279.2330	1.43	Tumor/Hyperplasia/Dysplasia > Healthy/Mastitis
Prostaglandin *	Fatty acid derivative	[M+Na-2H]^−^	389.1940	389.1946	1.54	Hyperplasia/Dysplasia > Tumor
PGP(C32:4-OH) *	Phospholipid	[M-2H]^2−^	404.1968	404.1969	0.25	Tumor/Mastitis > Hyperplasia/Dysplasia
NAT 20:4	Fatty acid derivative	[M+Cl]^−^	446.2122	446.2137	3.36	Hyperplasia/Dysplasia > Tumor
PGP(C38:6-3OH) *	Phospholipid	[M-2H]^2−^	460.2281	460.2341	13.04	Tumor/Hyperplasia/Dysplasia/Mastitis
CerP(d18:1/18:1)	Sphingolipid	[M-H]^−^	642.4850	642.4868	2.80	Mastitis
PA(C38:3) ^+^	Phospholipid	[M-H]^−^	725.5099	725.5127	3.86	Healthy > Tumor/Hyperplasia/Dysplasia/Mastitis
PE(C36:2) ^†^ PC(C34:2) ^†^	Phospholipid	[M-H]^−^ [M-CH^3^]^−^	742.5387	742.5392	0.67	Tumor > Healthy > Hyperplasia/Dysplasia/Mastitis
PE(C38:2) ^†^ PC(C36:2) ^†^	Phospholipid	[M-H]^−^ [M-CH^3^]^−^	770. 5705	770.5705	0.00	Healthy > Tumor/Hyperplasia/Dysplasia/Mastitis

The lipidomic profiles obtained represent univocal fingerprints of the analyzed tissues, and relative ion intensities were strictly connected to tissue type and correlated features. A comparison between healthy and pathological tissues (Figure 4) revealed the presence of key lipid species responsible for their differentiation.

Of note, relevant in normal mammary tissues are signals imputable to triglyceride (TG) species, detected in the range 890–950 *m*/*z* and identified on the basis of previous studies about cancer in both animals and humans [26,29] and confirmed by entering the molecular ions of more abundant isotopes into the online LipidMaps [33] and Human Metabolome Database [34] within a maximum mass error of 20 ppm, according to previous considerations about the mass accuracy of the iKnife instrumental setup [35].

The TG abundance in healthy tissues is noticeably counterbalanced by the low presence of certain glycerophospholipid (PLs) species (390–890 *m*/*z* range), which are predominant in neoplastic samples. Fatty acids (FA) and derived compounds, detected as deprotonated forms [M-H]^−^ in the mass range 250–350 *m*/*z*, are responsible for the most intense signals revealed in all spectra.

In order to reveal possible biomarkers of specific pathological conditions, the loading plots of the first two principal components (PC1 and PC2) were carefully examined and shown in Figure 5 and Figure 6, respectively. Therefore, Table 1 and Table 2, which list the tentatively identified compounds for the ion species reported in these loading plots, contain the major compounds responsible for sample differentiation, also named discriminant features.

As shown in Figure 1, PC1 is responsible for the differentiation between healthy and pathological conditions and explains 30.02% of the total variance of the model. In Figure 5, the main compounds identified in normal mammary tissues were reported at negative values, corresponding to chloride adducts [M+Cl]^−^ of polyunsaturated TG, including ions *m*/*z* 891.71 (TG(C52:3)), *m*/*z* 893.73 (TG(C52:2)), *m*/*z* 917.73 (TG(C54:4)), *m*/*z* 919.74 (TG(C54:3)), 921.75 (TG(C54:2)); deprotonated form [M-H]^−^ of phosphatidylethanolamine plasmalogen *m*/*z* 726.52 (PE(O-C36:3); and the FAs *m*/*z* 253.21 (C16:1) and *m/z* 281.24 (C18:1). Conversely, molecular species detected in other classes of mammary samples were observed in two main distinct mass regions at positive values of PC1. In the range 250–450 *m*/*z*, relevant signals were registered for deprotonated palmitic acid (*m*/*z* 255.23), the main fatty acid synthase (FASN) product in *de novo* synthesis chain also used for internal lockmass correction, and *m*/*z* 283.26, *m*/*z* 303.23 ions imputable to stearic (C18:0) and arachidonic (C20:4) acids, respectively. The saturated FA C18:0 is the first product of elongase ELOVL1, the enzyme involved in the generation of long chain saturated FAs, while the omega-6 arachidonic acid is the most commonly identified FA in PL species expressed in several neoplasms [36,37], and a key intermediate in prostaglandins synthesis, promoter substances of inflammatory processes [38]. In the same mass region, another polyunsaturated FA, which is docosatetraenoic acid (C22:4) at *m*/*z* 331.26, was detected, as well as the ions at *m*/*z* 391.22, *m*/*z* 417.23, *m*/*z* 419.25 and *m*/*z* 439.22, putatively identified by matching with Human Metabolome Database as dehydrate species of the lysophosphatidic acids Lyso-PA(C16:0), Lyso-PA(C18:1), Lyso-PA(C18:0) and Lyso-PA(C20:4). In the mass range 570–890 *m*/*z*, a huge number of signals were ascribable to pathological status, being them at positive values of PC1. All of them were identified as PLs, with the exception of the signal at *m*/*z* 572.48, tentatively identified by Zhang et al. [31] in human papillary thyroid carcinoma as Cer(C34:1) and peak at *m*/*z* 687.54 attributed to sphingomyelin SM(d18:1/16:0). Then, PA(C34:1), PA(C36:2), PA(C38:4) and PA(C40:4) were tentatively assigned to *m*/*z* 673.48, 699.49, 723.49 and 751.53, respectively. They are more clearly visible from the expansion in the insert of Figure 5. The other PL species, presumably derived from PA, found at positive values of PC1 and enlarged in the same insert, mainly belong to the classes of phosphatidylcholines (PC) and PE, with the exception of a peak at *m*/*z* 403.25 and 737.49, identified as oxidized phosphatidylglycerol PG(C36:1-2OH) and PG(C32:0;O). The first one was detected as a double-charged (double-deprotonated) ion, probably due to the presence of two hydroxyl groups on the FA chains. Finally, the PC1 loading plot shows high signals for *m*/*z* 766.53, 794.56 and 885.54 were identified as PE(C38:4)/PC(36:4), PE(C40:4) and PI(C38:4), respectively, further confirmed by the literature works dealing with the molecular characterization of human breast cancer tissues through the iKnife equipment [28]. Like PC1, PC2 also enabled a net demarcation between physiological and other conditions. Furthermore, the PCA plot in Figure 1 highlights that both hyperplastic/dysplastic and neoplastic tissues are mainly placed at positive PC2 values, while samples of inflammatory tissues are quite spread along PC2. The loading plot in Figure 6 reports the molecular species responsible for this differentiation. Apart from some lipid components discussed above, as they are already relevant in the PC1 loading plot, additional compounds were identified in the PC2 plot and reported in Table 2.

As shown in Figure 1, PC1 is responsible for the differentiation between healthy and pathological conditions and explains 30.02% of the total variance of the model. In Figure 5, the main compounds identified in normal mammary tissues were reported at negative values, corresponding to chloride adducts [M+Cl]^−^ of polyunsaturated TG, including ions *m*/*z* 891.71 (TG(C52:3)), *m*/*z* 893.73 (TG(C52:2)), *m*/*z* 917.73 (TG(C54:4)), *m*/*z* 919.74 (TG(C54:3)), 921.75 (TG(C54:2)); deprotonated form [M-H]^−^ of phosphatidylethanolamine plasmalogen *m*/*z* 726.52 (PE(O-C36:3); and the FAs *m*/*z* 253.21 (C16:1) and *m/z* 281.24 (C18:1). Conversely, molecular species detected in other classes of mammary samples were observed in two main distinct mass regions at positive values of PC1. In the range 250–450 *m*/*z*, relevant signals were registered for deprotonated palmitic acid (*m*/*z* 255.23), the main fatty acid synthase (FASN) product in *de novo* synthesis chain also used for internal lockmass correction, and *m*/*z* 283.26, *m*/*z* 303.23 ions imputable to stearic (C18:0) and arachidonic (C20:4) acids, respectively. The saturated FA C18:0 is the first product of elongase ELOVL1, the enzyme involved in the generation of long chain saturated FAs, while the omega-6 arachidonic acid is the most commonly identified FA in PL species expressed in several neoplasms [36,37], and a key intermediate in prostaglandins synthesis, promoter substances of inflammatory processes [38]. In the same mass region, another polyunsaturated FA, which is docosatetraenoic acid (C22:4) at *m*/*z* 331.26, was detected, as well as the ions at *m*/*z* 391.22, *m*/*z* 417.23, *m*/*z* 419.25 and *m*/*z* 439.22, putatively identified by matching with Human Metabolome Database as dehydrate species of the lysophosphatidic acids Lyso-PA(C16:0), Lyso-PA(C18:1), Lyso-PA(C18:0) and Lyso-PA(C20:4). In the mass range 570–890 *m*/*z*, a huge number of signals were ascribable to pathological status, being them at positive values of PC1. All of them were identified as PLs, with the exception of the signal at *m/z* 572.48, tentatively identified by Zhang et al. [31] in human papillary thyroid carcinoma as Cer(C34:1) and peak at *m/z* 687.54 attributed to sphingomyelin SM(d18:1/16:0). Then, PA(C34:1), PA(C36:2), PA(C38:4) and PA(C40:4) were tentatively assigned to *m/z* 673.48, 699.49, 723.49 and 751.53, respectively. They are more clearly visible from the expansion in the insert of Figure 5. The other PL species, presumably derived from PA, found at positive values of PC1 and enlarged in the same insert, mainly belong to the classes of phosphatidylcholines (PC) and PE, with the exception of a peak at *m/z* 403.25 and 737.49, identified as oxidized phosphatidylglycerol PG(C36:1-2OH) and PG(C32:0;O). The first one was detected as a double-charged (double-deprotonated) ion, probably due to the presence of two hydroxyl groups on the FA chains. Finally, the PC1 loading plot shows high signals for *m/z* 766.53, 794.56 and 885.54 were identified as PE(C38:4)/PC(36:4), PE(C40:4) and PI(C38:4), respectively, further confirmed by the literature works dealing with the molecular characterization of human breast cancer tissues through the iKnife equipment [28]. Like PC1, PC2 also enabled a net demarcation between physiological and other conditions. Furthermore, the PCA plot in Figure 1 highlights that both hyperplastic/dysplastic and neoplastic tissues are mainly placed at positive PC2 values, while samples of inflammatory tissues are quite spread along PC2. The loading plot in Figure 6 reports the molecular species responsible for this differentiation. Apart from some lipid components discussed above, as they are already relevant in the PC1 loading plot, additional compounds were identified in the PC2 plot and reported in Table 2. 

Specifically, at negative values of PC2, where healthy tissues are located, FAs can be identified in the range 250–305 *m*/*z*, and among them, the polyunsaturated linoleic acid (C18:2) at *m*/*z* 279.23 was not present within PC1 features, thus playing a minor role in the discrimination compared to saturated and monounsaturated FAs, as well as with respect to arachidonic acid. At negative PC2 values, a signal at *m*/*z* 642.48 also appeared, and it was putatively assigned to CerP(d18:1/18:1) (ceramide phosphate class) by matching the accurate mass to the Lipid Maps database. In agreement with the PCA plot in Figure 1 and the spectral comparison in Figure 6, it was mainly identified in inflamed tissue samples. Some less abundant PLs at *m*/*z* 725.50, 742.53 and 770.57 were also present among the PC2 discriminant features (Figure 6 and Table 2), while they were not visible in the PC1 loading plot. They were present in the MS spectra of all the samples, but differences in their ratio can represent a discriminant factor. On the other hand, at positive PC2 values, two FA derivates were mainly detected in hyperplastic/dysplastic tissues. Specifically, the signal at *m*/*z* 389.19 was tentatively identified as oxidation products of arachidonic acid, known as prostaglandins, and clearly related to disease conditions. The signal at *m*/*z* 446.21 was also identified as an arachidonic acid derivate, i.e., N-arachidonoyl taurine (NAT). Finally, two oxidized PG species were assigned for *m*/*z* 404.19 and 460.22. As the oxidized PG(C36:1-2OH) identified in the PC1 loading plot, they represent signaling molecules in the inflammatory process. Compared to PG(C36:1-2OH), they are polyunsaturated lipids, which presumably contain oxidized products of the arachidonic acid.

## 3. Discussion

In the present study, a lipidomics-based method was applied to metabolic phenotyping canine mammary tissues, investigating lipid profiles in the normal mammary gland and comparing changes in lipid species in hyperplastic/dysplastic, inflammatory and cancerous mammary tissues. The obtained lipidomic profiles represented unique fingerprints of the analyzed tissues, and the relative ion intensities were closely related to the type of tissue, reflecting the different histological classifications of samples. Furthermore, the main discriminating molecules were identified to distinguish between healthy and neoplastic tissues, making these molecules promising diagnostic biomarkers. The data obtained suggest that REIMS technology may be useful to accurately differentiate canine mammary tissues based on cellular lipid constituents, allowing comparing changes in cellular metabolism with alterations in tissue morphology.

First, the purpose of the iKnife application was the construction of an internal database with spectra obtained by analyzing a representative number of histologically classified mammary samples. Starting from the concept of machine learning, the collection of mass spectra and, therefore, the construction of an extensive database is a mandatory step that makes the system able to learn from the acquired data and subsequently identify specific patterns, returning prediction results about unknown samples. Therefore, an adequate number of samples is required for a meaningful statistical analysis, viz., to compare typical lipid profiles of physiological and major pathological conditions affecting the canine mammary gland.

In this study, with the exception of the normal mammary gland tissues, all included samples appeared as neoformations on clinical examination and, therefore, surgically removed. The different diagnostic and prognostic significance for mammary masses were further established based on histological examination.

Acquired MS spectra were used to build a classification model able to discriminate between mammary samples based on different histopathological diagnoses, using two different mathematical approaches, and the model was further validated for classification accuracy by performing two different *in silico* tests. The REIMS profiles were classified with an overall accuracy of 94.76% and 90.05% using PCA-LDA analyses, with a very low percentage of failures and outliers. Furthermore, tumors were never misidentified as healthy mammary tissue or inflammatory samples, and significant differences in lipid profiles were mainly observed between normal tissues and CMT, reflecting significant differences in lipid metabolism. Furthermore, although the group of CMT represented the one with the highest variability in terms of morphology, as this group included both benign and malignant tumors with different grades of malignancy, the supervised LDA analysis reduced the differences among samples, allowing identifying all neoplastic tissues in the same class correctly. Interestingly, the recognition capability of REIMS was further tested on unknown mammary samples with known histological diagnoses, allowing obtaining the output of the analysis in near real-time, and all samples were correctly identified with a 98% similarity rate with spectra present in the database.

The identification of lipid species in our samples, based on correspondence with those available in the LipidMaps online library, revealed significant differences, especially between normal and cancerous mammary samples, allowing us to further identify the main “discriminant molecules”. Notably, triglyceride identification was increased in normal mammary samples, while the abundance of ions associated with PL species was increased in both benign and malignant tumors, in which the intensity of ions associated with triglyceride species was reduced.

Interestingly, lipid analyses of breast cancer tissue samples showed an increase in PL content compared to non-cancerous healthy breast tissue [39], demonstrated using ambient MS-based lipidomic in breast cancer [26]. PLs are essential components of all cell membranes, and their content has been shown to increase with tumor cell transformation and tumor progression [40]. Furthermore, the high content of PLs in tumors was inversely correlated with patient survival and also related to resistance to chemotherapies [41].

Cancer cell metabolism promotes lipid synthesis of FAs by anaerobic glycolysis, producing energy and pyruvate (the so-called “Warburg effect”), and the activation of lipid metabolism has been recognized as a hallmark of tumorigenesis in breast cancer cells [40]. Of note, the involvement of *de novo* FAs in the PL biosynthesis rather than TGs is a key point responsible for cell proliferation since PLs represent the main constituents of cell membranes [42]. In particular, one of the first steps in the lipidomic pathway is the formation of PAs from Lyso-PAs, and FAs are incorporated into complex lipids via PAs as intermediate metabolites [43]. Evidence of the role of Lyso-PLs on the growth regulation of neoplastic cells and the manipulation of the immune system supports their presence, while PAs also act as precursors for *de novo* generated PLs or those existing and can serve as cell survival signaling molecules [43]. Although *de novo* FAs synthesis incorporated into membrane PLs usually increases during cancer progression, few healthy tissues as adipose tissue, which is an important component of the breast tissue, use the same mechanism to generate FAs as reported elsewhere [32]. However, differences in lipid saturation between healthy and cancerous mammary tissues were observed in our study. In particular, a greater expression of saturated FAs was observed in mammary tumors, while unsaturated FAs were identified in the healthy mammary gland. One of the advantages of the high rate of de novo lipogenesis in cancer cells is the synthesis of saturated and monounsaturated FAs, which are more stable than polyunsaturated FAs; consequently, cancer cells are less susceptible to chemotherapy-induced oxidative stress [44].

In agreement with other studies, the PLs identified and most expressed in compromised tissues were PEs (PCs) and PAs, particularly in tumors [21,45]. Concentrations of the two major PL components, phosphatidylethanolamine (PE) and phosphatidylcholine (PC), increased with increasing breast cancer grade, indicating that the rate of PL synthesis increases with oncogenesis and tumor progression compared to normal tissue [40]. In particular, high PEs synthesis was demonstrated as a common metabolic adaptation strategy to the stress of breast cancer cells [46]. Of note, PC is one of the main lipids in mitochondria, which represent crucial organelles for cellular adaptation to metabolic insults [47]. In particular, PCs are generally the most abundant PL species in mammalian cells. PC synthesis and metabolism in cancer progression have been investigated and PC species have been proposed as putative diagnostic markers and therapeutic agents in breast cancer [32,48].

Signals related to sphingolipids, including sphingomyelin and ceramide, as well as oxidized lipids, including oxidized PG, were also identified in pathological tissues analyzed here. Sphingolipids are essential constituents of cell membranes and play crucial roles in cell homeostasis, as well as in the onset and progression of several diseases [49]. In particular, sphingomyelin is a known regulator of cell proliferation and apoptosis in neoplastic and inflammatory processes [24,50], while ceramide induces cell cycle arrest, regulating cell survival and promoting apoptosis, and plays a central role in the regulation of inflammatory responses [49]. The phosphorylation of ceramide produces ceramide 1-phosphate, an important metabolite with mitogenic and prosurvival properties, also involved in the stimulation of the inflammatory response, the release of arachidonic acid and in the formation of prostaglandins through the activation of phospholipase enzymes [49,51]. In our study, the identification of sphingomyelin and ceramide in the PC1 loading plot further validates the presence of ceramide phosphates since the latter are biosynthetic products of the former. Finally, the identification of oxidized lipids here may be related to the presence of possible cellular alterations responsible for the apoptotic mechanism [52], although it was recently shown that the oxidation of FAs protects neoplastic cells from apoptosis in triple-negative breast cancer by increasing the synthesis of PLs and lipids of the mitochondrial membranes, thus, counteracting the mitochondrial apoptotic pathways [53].

Finally, regarding the NAT detected in hyperplastic/dysplastic mammary tissues, it was probably derived from an endogenous synthesis activated in the presence of high levels of arachidonic acid, released from PLs by the action of phospholipase activated under stress conditions [54]. Such compounds have been shown to exert an anti-proliferative action against cancer cells [55]. Furthermore, omega-6 arachidonic acid is the most commonly identified FA in PL species expressed in several neoplasms [36,37] and a key intermediate in the synthesis of prostaglandins, substances that promote inflammatory processes [38].

## 4. Materials and Methods

### 4.1. Samples and Sample Treatment

Mammary tissue samples analyzed in this study were surgically removed from a cohort of female dogs referred to private veterinary clinics in the province of Messina and Catania (Sicily; Southern Italy). Dogs underwent conservative surgery (i.e., partial or radical mastectomy) as the only treatment, and the owners provided consent with curative intent. Healthy mammary tissues were sampled from dogs that underwent routine necropsies at the Department of Veterinary Sciences of the University of Messina. All collected samples were specularly sectioned and either fixed in 10% neutral buffered formalin to be routinely processed for histology and stored at −80 °C for iKnife analyses at the Department of Chemical, Biological, Pharmaceutical and Environmental Sciences of the University of Messina. The intrinsic hydration of some tissues allowed performing their direct sampling by the electrosurgical unit without any sample pretreatment or modification.

### 4.2. Histopathology

Three-micrometer thick sections from formalin-fixed, paraffin-embedded tissues were stained with haematoxylin-eosin (HE) for histological examination. Histological classification was based on criteria defined by Zappulli et al. [56], and histological grading was performed according to Peña et al. [57], based on the histological features of tubule formation, nuclear pleomorphism and mitotic count.

### 4.3. iKnife Analysis

#### 4.3.1. Chemicals

HPLC-MS grade methanol, water and 2-propanol (IPA) were purchased from Merck life Science (Merck KGaA, Darmstadt, Germany) and employed for Venturi pump cleaning and MS analysis. LC-MS grade formic acid, sodium hydroxide and Leucine encephalin standard (LeuEnk) were provided by Waters Corporation (Wilmslow, UK) and used to perform the setup and calibration of a high-resolution Q-TOF detector.

#### 4.3.2. iKnife Instrumentation and Analytical Conditions 

A surgical diathermy system (Erbe VIO 50 C, Tuebingen, Germany) composed of an electric current generator and coupled to a monopolar handpiece was employed to perform tissue sampling, which involves the vaporization of cellular constituents by the Joule effect. Particularly, dry-cut (DC) mode and 30 W frequency were applied to power the electrosurgical knife, thus heating its metal end; then, the cutting led to the release of very informative molecular vapors. These products were aspirated via a 4 m × 4.11 mm o.d., 2.53 mm i.d., PTFE tube to the inlet of the REIMS (Rapid Evaporative Ionization Mass Spectrometry) source (Waters Corporation, Wilmslow, UK) for the ion formation as a result of a collision with a helically coiled surface heated by a constant electric current set to 4.5 A and 4.2 V (Kanthal D 1.0 × 0.1 mm). In order to avoid carryover phenomena between subsequent analyses, the source cleanup and promotion of molecular ionization were guaranteed by introducing a constant flow of IPA (0.05 mL/min) into the source through an automated Harvard Apparatus equipped with a syringe pump (Trajan Scientific, Crownhill, UK, 10 mL). The REIMS source was installed on a Xevo G2 XS Q-Tof (Waters Corporation, Wilmslow, UK). All tissues were analyzed in sensitivity mode at a resolution of about 20,000 FWHM (Full Width at Half Maximum). The MS spectra were acquired in negative ionization mode in the mass range 100–1200 *m*/*z* at a scan rate of 0.5 s. The electrosurgical knife blade, the PTFE evacuation transfer tube and the individual components of the Venturi tube were cleaned in methanol MS grade after every 10 analyses.

#### 4.3.3. Data Processing and Statistical Analysis 

Acquisition of REIMS-Tof-MS analyses and data pretreatment were performed by using the software package Masslynx version. 4.1 (Waters Corporation, Wilmslow, UK). Raw files were processed for background subtraction (e.g., removal of electronic noise), normalization and accurate mass correction of acquired ions using the endogenous matrix 255.233 *m*/*z* that corresponds to the deprotonated molecule of palmitic acid (C_16_H_31_O_2_). This process is necessary to make different analyses, even performing in different periods, fully comparable. The processed MS data were imported into a spectral library through the LiveID™ software version 1.2 (Waters Corporation, Wilmslow, UK) to be used for chemometric analysis in order to reveal similarities and differences between individual tissues. In particular, a multivariate statistical model was built to achieve discrimination between the samples. Hence, the model was validated and tested for the recognition of unknown samples. Specifically, the statistical analysis implied the use of two different mathematical approaches applied sequentially by the software and consisting of PCA and LDA. The first method, defined as “unsupervised”, places each MS spectrum into a well-defined position within a multidimensional space, regardless of tissue type (or class) to which it belongs. On the other hand, LDA or “supervised” approach was employed as a classification technique useful for the building of predictive models (maximizing the inter-class variance and minimizing the intra-class one) [58], which allows classifying MS spectra of unknown samples as belonging to a specific tissue type during the recognition step. In order to evaluate the robustness of the resulting PCA/LDA statistical model and define its predictive capability, a 5-fold stratified validation test was performed. It works by dividing the data set used for the statistical model into five different partitions, each of which contains a representative proportion of each class. Four of these partitions are used to build a statistical model with the same parameters as the original one and used to classify the remaining portion of data not included. This approach is repeated for the other 5 cycles, and the excluded partition every time undergoes the classification process of the new model built with the remaining four. LiveID software returns a detailed report at the end of the validation test that includes data classified as correct, incorrect (failures) and those defined as anomalous (outliers). The validated model was finally used for the recognition in playback (in a phase subsequent to the acquisition of the MS spectra), and in real-time (at the same time), of unknown data not included in the statistical model.

## 5. Conclusions

In conclusion, the present work aimed to determine the discriminant features between normal mammary glands and mammary samples with different pathological lesions. Compared to most of the previous work related to the differentiation of biological tissues, a more in-depth elucidation of these discriminating features was carried out since a wider mass range was investigated. In particular, only the PL region (600–900 *m*/*z*) was normally monitored, while the present research highlighted the possibility of detecting additional marker compounds, such as FA derivatives, oxidized lipids and Lyso-PA, here tentatively identified for the first time in this type of samples. The data obtained here suggest that the iKnife technique is able to accurately profile mammary tissues based on cellular chemical constituents, allowing to compare chemical changes in cellular metabolism with tissue morphology, suggesting that the iKnife could be used as an accessory diagnostic tool intraoperatively in the future. However, further larger research cohorts of ex vivo samples, as well as adequate translation of the ex vivo recognition software for intraoperative use, are required to validate this technique and before conclusions can be drawn on intraoperative diagnostic accuracy

## Figures and Tables

**Figure 1 ijms-23-10562-f001:**
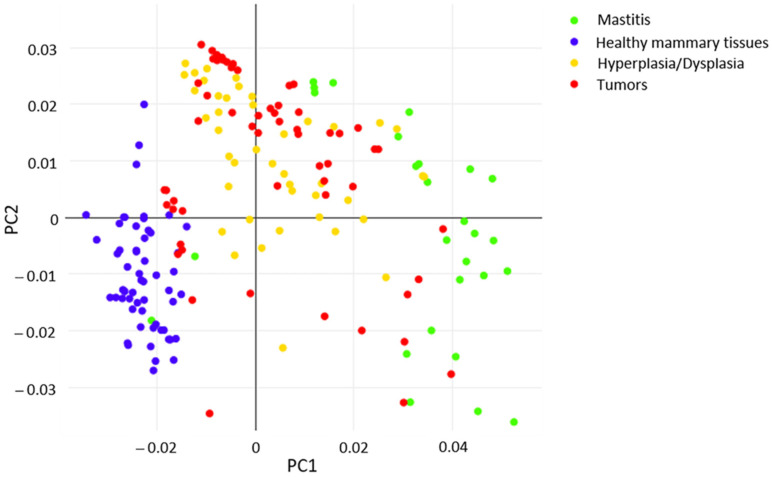
PCA score plot of healthy and pathological status included in the model.

**Figure 2 ijms-23-10562-f002:**
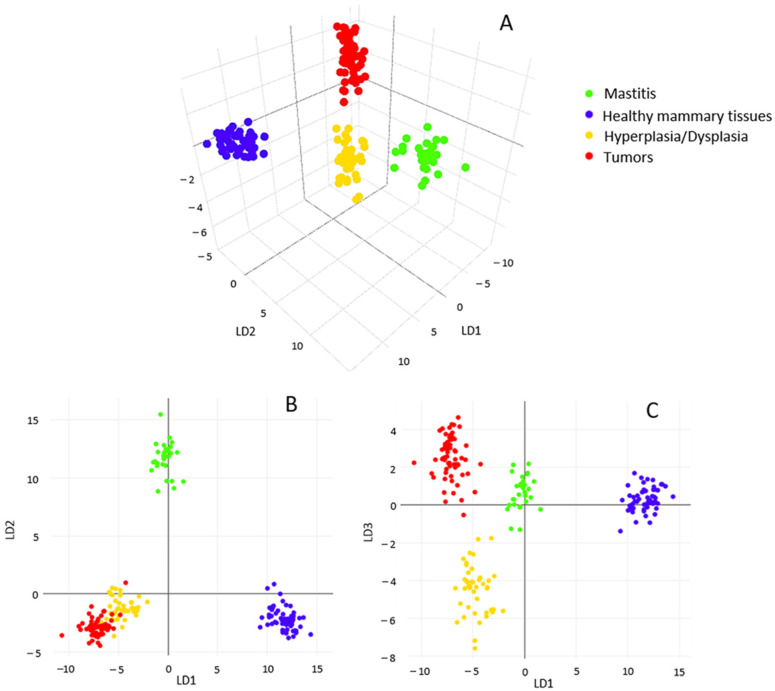
Tridimensional visualization of PCA/LDA statistical model (**A**); bidimensional score lots along first three main linear discriminants (LD): LD1/LD2 (**B**) and LD1/LD3 (**C**).

**Figure 3 ijms-23-10562-f003:**
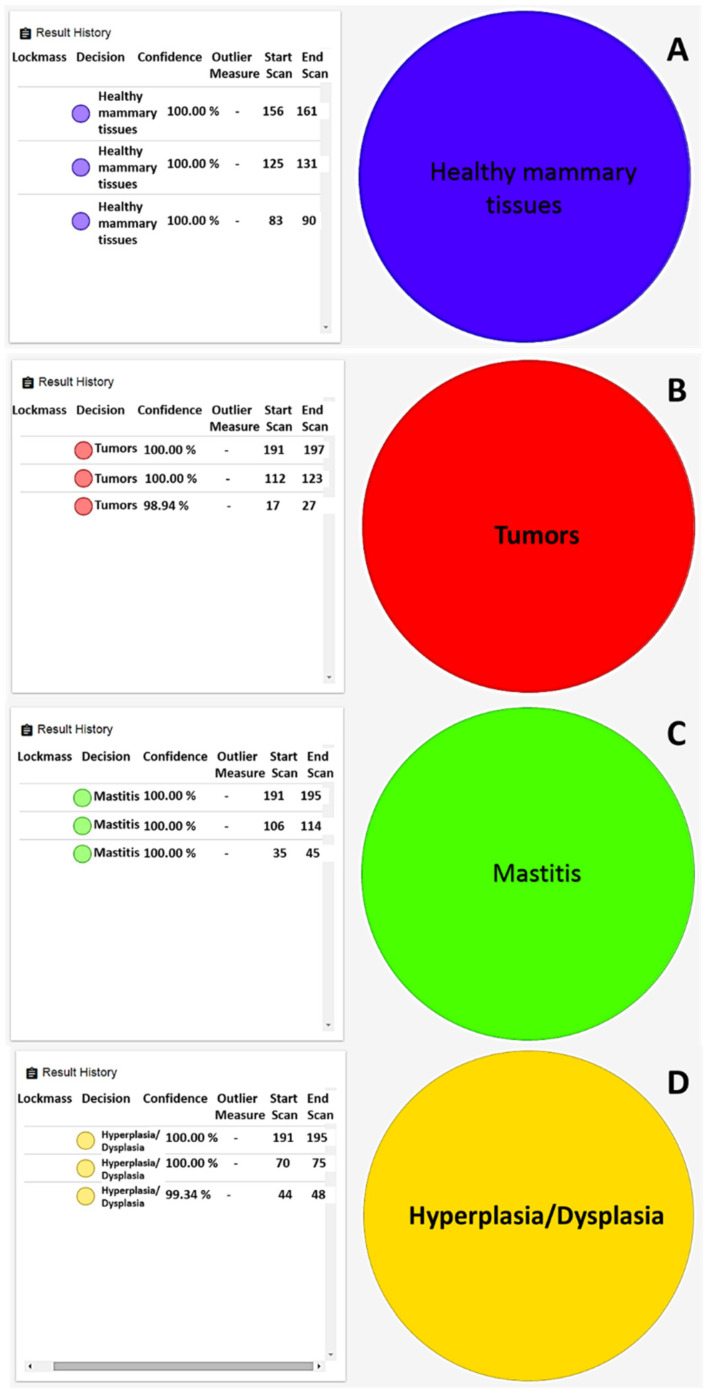
Real-time identification of (**A**) healthy mammary gland, (**B**) comedocarcinoma as malignant tumor, (**C**) margin tissue of comedocarcinoma, (**D**) ductal ectasia.

**Figure 4 ijms-23-10562-f004:**
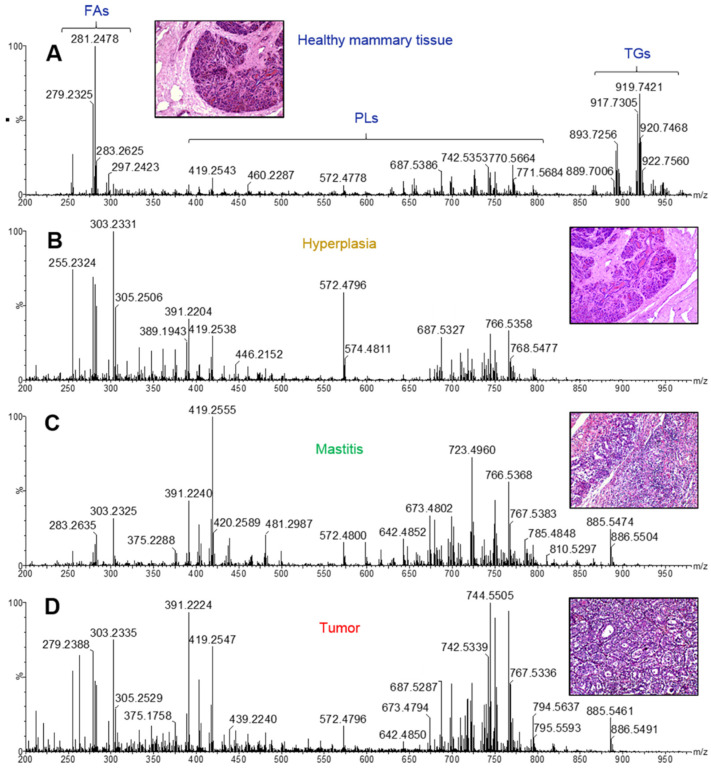
Representative REIMS(-) spectra of (**A**) healthy mammary gland, (**B**) hyperplastic/dysplastic tissue, (**C**) mastitis, (**D**) tumor.

**Figure 5 ijms-23-10562-f005:**
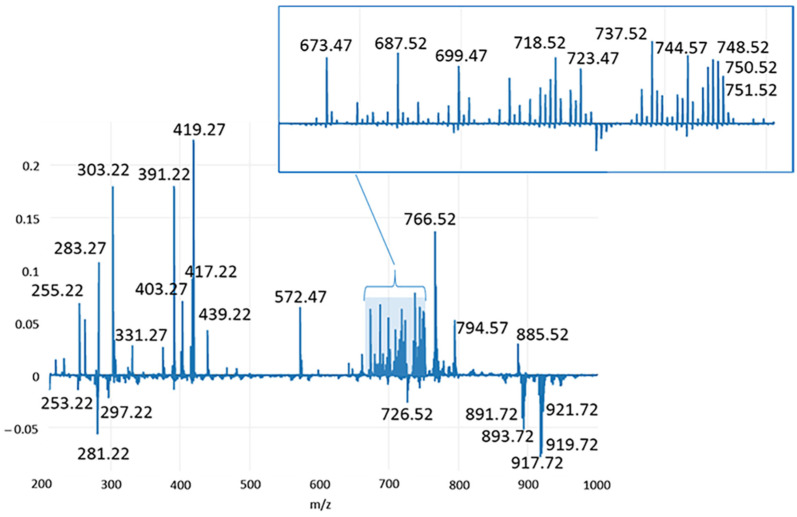
Loading plot relative to the PC1 component, explaining the 30.02% of the total variance of the model.

**Figure 6 ijms-23-10562-f006:**
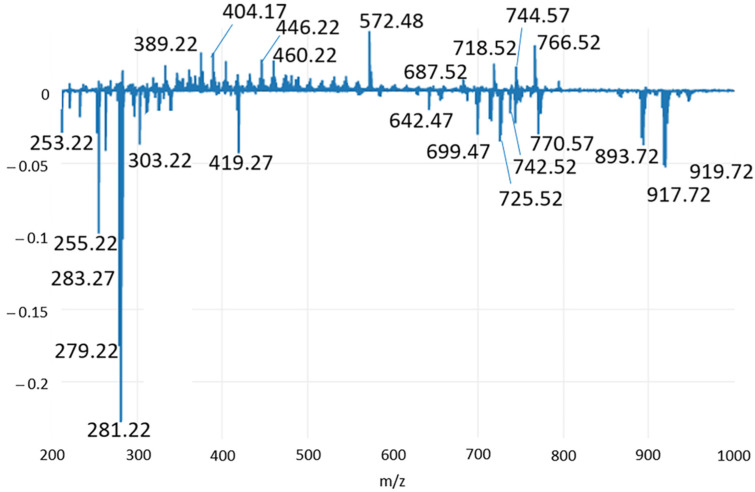
Loading plot relative to the PC2 component, explaining the 30.02% of the total variance of the model.

## Data Availability

Not applicable.

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
