# Peer review of "Rapid Evaporative Ionization Mass Spectrometry-Based Lipidomics for Identification of Canine Mammary Pathology"

_ijms, 2022, doi:10.3390/ijms231810562_

Round 1
Reviewer 1 Report
The manuscript entitled ‘Rapid Evaporative Ionization Mass Spectrometry-based Lipidomic for Identification of Canine Mammary Pathology’ by Mangraviti et al describes the use of the iKnife tool to distinguish healthy canine mammary tissue from cancerous or otherwise pathological tissue samples based on lipid profiling. While the premise of the paper is acceptable and the PCA/LDA demonstrating the separation of the datasets looks convincing, insufficient care went into the preparation of the manuscript and I cannot properly gauge its scientific soundness or recommend its publication in its current form. Specific comments follow.
The overall readability of the manuscript needs to be improved. Both the Introduction and the Discussion sections are a single paragraph each (the latter a full two pages long!) – I find the text impossible to follow as is and so making any judgements on the scientific soundness of the paper’s contents is difficult. Please rewrite and break this up into smaller paragraphs.
English language editing may be needed – the title has a typo which ruins the first impression of the manuscript a little; consider ‘lipidomics’ or ‘lipidomic analysis’ instead of ‘lipidomic.’
Figure 3 – I understand these are basically screenshots of the software. However, the text in the left-hand panels is much too small to be legible, and needs to be altered for publication.
Line 113 – healthy sample datapoints are described alternately as purple or blue. Please choose one to avoid confusion.
Line 141, ‘an equal number’ – what number exactly?
Line 157, ‘in the mass range 100-200 m/z’ – this appears to be incorrect. Please fix.
Table 1 – what does ‘confirmed by literature’ mean exactly? Also, what exact method/software was used to assign the peaks to specific lipid species? Was the determination based solely on exact mass?
On a related note – how selective is this process for lipids specifically? Are the parameters used for iKnife/mass spec operation optimized to favor ionization of lipids over other analytes? What proportion of peaks in the loading plots could be attributed to lipids, versus other classes of metabolites (alternatively, what proportion of peaks could not be confidently identified)?
The supplementary information file is missing (the one given is a copy of the manuscript).
Author Response
Dear Reviewer,
Please find attached our responses regarding the manuscript. We have addressed all of your concerns and all corrections have been tracked in the manuscript.

Reviewer 2 Report
The manuscript numbered ijms-1886953 deals with the utility of iKnife technology in metabolic profiling and differentiation of canine mammary tissues (in pathological and physiological state) on the basis of the histologically validated ex vivo samples.
This paper may be of utmost interest for the journal’ audience. Manuscript raise the most recent scientific and practical matters and provide proper explanation of them. The paper is well written, the need of the research is enough justified, experiment was well thought out, planned and executed. Materials and methods are exhaustively described, used methods are novel and well-chosen to achieve the main aim of the study. Obtained results were sufficiently described discussed, discussion are explanatory and informative. Special emphasis should be put on the fact that in this study clinical, methodological and data science approaches were combined and successfully applied to the real cases. Concise conclusion sum up the most important achievements of this paper and also provide a starting point for further, deepened research with use of this analytical platform.
Author Response
Dear Reviewer,
Please find attached our responses to comments regarding the manuscript.

Round 2
Reviewer 1 Report
My main concerns have been satisfied.
Just a minor point that escaped me on the first review round - please ensure that H20 is corrected to H2O in table 2 prior to publication.